# Efficacy of Glecaprevir/Pibrentasvir for Real-World HCV Infected Patients in the Northern Part of Tokyo, Japan

**DOI:** 10.3390/jcm10235529

**Published:** 2021-11-26

**Authors:** Yoichiro Yamana, Tatsuo Kanda, Naoki Matsumoto, Masayuki Honda, Mariko Kumagawa, Reina Sasaki, Shini Kanezawa, Taku Mizutani, Hiroaki Yamagami, Ryota Masuzaki, Tomotaka Ishii, Kazushige Nirei, Mitsuhiko Moriyama

**Affiliations:** Division of Gastroenterology and Hepatology, Department of Medicine, Nihon University School of Medicine, 30-1 Oyaguchi-kamicho, Itabashi-ku, Tokyo 173-8610, Japan; yamana.yoichiro@nihon-u.ac.jp (Y.Y.); matsumoto.naoki@nihon-u.ac.jp (N.M.); honda.masayuki@nihon-u.ac.jp (M.H.); kumagawa.mariko@nihon-u.ac.jp (M.K.); sasaki.reina@nihon-u.ac.jp (R.S.); kanezawa.shini@nihon-u.ac.jp (S.K.); mizutani.taku@nihon-u.ac.jp (T.M.); yamagami.hiroaki@nihon-u.ac.jp (H.Y.); masuzaki.ryota@nihon-u.ac.jp (R.M.); ishii.tomotaka@nihon-u.ac.jp (T.I.); nirei.kazushige@nihon-u.ac.jp (K.N.); moriyama.mitsuhiko@nihon-u.ac.jp (M.M.)

**Keywords:** chronic kidney disease, DAA failure, hemodialysis, HCV, NS5A P32 deletion mutant

## Abstract

Hepatis virus C (HCV) infection causes liver cirrhosis and hepatocellular carcinoma (HCC) worldwide. The objective of our study was to examine the effects of the HCV nonstructural protein (NS) 3/4A inhibitor glecaprevir/NS5A inhibitor pibrentasvir on real-world HCV patients in the northern part of Tokyo, Japan. Although 106 patients were consecutively included, a total of 102 HCV-infected patients with chronic hepatitis or compensated cirrhosis, who received 8- or 12-week combination treatment with glecaprevir/pibrentasvir and were followed up to week 12 after the end of treatment were analyzed retrospectively. Only three patients discontinued treatment due to adverse events; however, they achieved a sustained virologic response at 12 weeks (SVR12). Finally, SVR rates were 99.0% (101/102). Only one patient without liver cirrhosis was a treatment relapser who received hepatic resection for HCC approximately two years after commencement of the 8-week combination treatment with glecaprevir/pibrentasvir. After the exclusion of patients with HCV genotype 1b and P32 deletion in the HCV NS5A region, a 12-week combination of glecaprevir/pibrentasvir led to SVR12 in all nine direct-acting antiviral-experienced patients. Glecaprevir/pibrentasvir had a high efficacy and an acceptable safety profile for real-world HCV patients in a single hospital in Japan.

## 1. Introduction

Chronic hepatitis C virus (HCV) infection causes liver cirrhosis and hepatocellular carcinoma (HCC), which are life-threatening diseases worldwide [1,2]. The 12-week combination treatment of direct-acting antivirals (DAAs) has higher eradication rates of HCV (ranging from 95–100%) with fewer adverse events [3,4]. However, there were still some non-responders, who needed other therapeutic regimens [5,6].

Glecaprevir and pibrentasvir are inhibitors of HCV nonstructural (NS) protein 3/4A protease and NS5A, respectively [7]. These combinations of glecaprevir/pibrentasvir have pangenotypic anti-HCV activity with a high barrier to resistance, primarily biliary excretion and negligible renal excretion [8,9,10]. Therefore, the combination of glecaprevir/pibrentasvir could be used for the treatment of HCV-infected patients on dialysis and those with severe renal impairment or for the retreatment of HCV-infected patients with previous DAA treatment failure.

We report here the real-world experience with glecaprevir/pibrentasvir from the northern part of Tokyo, Japan, generated from a retrospective study of the effectiveness and safety of an 8- or 12-week course of treatment with glecaprevir/pibrentasvir for HCV-infected patients with chronic hepatitis or compensated cirrhosis in daily clinical practice. We emphasize the efficacy of this regimen in the Japanese population while also highlighting the safety profile.

## 2. Patients and Methods

### 2.1. Study Design and Patients

This retrospective study enrolled patients with chronic HCV infection who started to receive interferon-free combination treatment with glecaprevir/pibrentasvir from 1 November 2017 to 31 August 2019. A total of 106 patients were initially included. Eligible patients were 20 years of age and older and had chronic hepatitis or compensated cirrhosis (Child-Pugh A cirrhosis). An 8- or 12-week combination treatment of glecaprevir/pibrentasvir was given in DAA-naïve patients, and a 12-week combination treatment of glecaprevir/pibrentasvir was given in DAA-experienced patients (Figure 1).

The exclusion criteria were as follows: (1) Child-Pugh B or C cirrhosis; (2) serious other medical conditions such as severe anemia, pulmonary diseases, or heart diseases; (3) the presence of active hepatocellular carcinoma (HCC); (4) human immunodeficiency infection; and (5) patients with virologic failure who had both HCV genotype 1b infection and P32 deletion in the HCV NS5A region at baseline [11]. Patients on dialysis and those with severe chronic kidney disease (CKD) or those with a history of curative HCC treatment were included. Some of these patients had been included in other studies [6,12].

The protocol of this single center study followed the Declaration of Helsinki. The ethics committee of Nihon University School of Medicine Itabashi Hospital approved this retrospective study (protocol number RK-181009-04, and RK-180911-12). Participation in the study was posted at the website of our institution, and informed consent was obtained from all patients.

### 2.2. Serum Biochemical Tests and Hematological Tests

Serum biochemical tests including liver function tests and the estimated glomerular filtration rate (eGFR), and hematological tests were performed according to standard methods [6].

### 2.3. Measurement of HCV RNA Levels and Determination of HCV Genotypes

Serum HCV RNA levels were measured by COBAS TaqMan assay (Roch Diagnostics, Tokyo, Japan) with detection limits of ~1.2 LIU/mL. At least, HCV RNA levels were determined at pre-treatment, at the end of treatment and after 12 weeks at the end of treatment. SVR12 was used as the SVR to evaluate the virological response. Virus clearance was defined as undetectable HCV RNA. HCV genotypes were determined by the antibody serotyping method [13] or PCR-based assay with genotype-specific PCR primers [14]. In a non-SVR patient, HCV NS5A resistance-associated substitutions (RASs) at 31L and 93Y were determined by a commercially available direct-sequencing assays (SRL Laboratory, Tokyo, Japan) [15].

### 2.4. Assessment of Advanced Liver Fibrosis and Diagnosis of Cirrhosis and HCC

Ultrasonography and hepatic transient elastography on a FibroScan 502 with an M probe (Echosens, Paris, France) were performed. In general, liver stiffness equal to or more than 12.0 kPa or the sign of cirrhosis was considered liver cirrhosis. The sign of cirrhosis was the existence of varices in the esophagus and/or stomach on upper gastrointestinal endoscopy or the existence of compatible findings of liver cirrhosis in computed tomography (CT) scanning or magnetic resonance imaging (MRI). In this study, Child-Pugh A cirrhosis was defined as compensated cirrhosis. In general, HCV-infected patients were followed up through an HCC surveillance program based on ultrasonography evaluations with or without tumor markers/CT/MRI at least every six months [6].

### 2.5. Statistical Analysis

Data are expressed as the mean ± standard deviation (SD). Statistical analysis was performed by the Student’s *t*-test or chi-squared test. A *p*-value < 0.05 was considered a statistically significant difference.

## 3. Results

### 3.1. Patients’ Characteristics

A total of 106 consecutive HCV-infected patients who commenced an 8- or 12-week combination treatment of glecaprevir (300 mg daily)/pibrentasvir (120 mg daily) (fixed-dose compound: Maviret, AbbVie, Tokyo Japan) were initially included (Figure 1 and Figure 2).

Of them, 103 patients completed the treatment (Figure 2). As 4 patients were lost to follow-up, 99 patients were included in the study. A total of 3 out of the 106 (2.8%) patients who discontinued treatment by adverse events were also included in the present study, as the SVR was judged in these three patients. In total, 102 patients aged older than 20 years in whom sustained virologic response (SVR) at 12 weeks after the end of treatment (SVR12) was judged were defined as eligible (Figure 2). Overall, 102 patients were included in this retrospective analysis (Table 1).

The characteristics of the 102 patients at baseline are shown in Table 1. In total, 81 were treatment-naïve patients and did not receive any interferon-including or DAA-including regimens. The HCV subgenotypes of 102 patients were as follows: 1a:1b:1 unknown subgenotype: 2a:2b:2 unknown subgenotype: 3a:2:51:1:24:19:2:3. Six patients went on artificial dialysis for chronic kidney failure.

### 3.2. The Efficacy and Safety of the 8- or 12-Week Combination Treatment of Glecaprevir/Pibrentasvir

A total 99 patients completed the treatment, and 3 patients discontinued the treatment due to severe adverse events. Among these three patients, an 85-year-old female patient with HCV genotype 1b and chronic hepatitis, stopped the treatment due to her cerebral hemorrhage at 4 weeks after the commencement of the treatment and achieved SVR12; a 74-year-old female patient with HCV genotype 1b and cirrhosis, stopped the treatment due to her hyperbilirubinemia (total bilirubin, 3.8 mg/dL; direct bilirubin, 2.5 mg/dL) at 6 weeks after the commencement of the treatment and achieved SVR12; and a 63-year-old male patient with HCV genotype 1b and cirrhosis, stopped the treatment due to his hyperbilirubinemia (total bilirubin, 3.9 mg/dL; direct bilirubin, 2.8 mg/dL) at 8 weeks after the commencement of the treatment and achieved SVR12. All these patients had diabetes mellitus, and two patients possessing hyperbilirubinemia had cirrhosis. Finally, except for only 1 patient, 101 patients achieved SVR.

Among the three HCV genotype 3a-infected patients, one and two patients were treated with 8- and 12-week combination treatment of glecaprevir/pibrentasvir, respectively, and all three patients achieved SVR12. Among the 24 patients with compensated cirrhosis after excluding two patients who discontinued the treatment due to adverse events, 6 and 18 patients were treated with 8- and 12-week combination treatment of glecaprevir/pibrentasvir, respectively, and all 24 patients achieved SVR12. Among the 73 patients with chronic hepatitis after excluding one patient who discontinued the treatment due to adverse events, 64 and 9 patients were treated with 8- and 12-week combination treatment of glecaprevir/pibrentasvir, respectively, and 72 patients (98.6%) achieved SVR12.

The characteristics of one relapse patient is shown in Table 2. In this patient, HCV RNA was relapsed after 12 weeks of the end of treatment. At this time, his HCV RNA level was 5.2 LIU/mL. He had stopped coming to our outpatient clinic due to his circumstance for two years. He received hepatic resection for HCC ~2 years after the commencement of 8-week combination treatment of glecaprevir/pibrentasvir. HCV RNA level was 5.5 LIU/mL before his surgery. Histological evaluation of non-HCC liver revealed no existence of liver cirrhosis. Due to his severe heart disease, he was retreated with the 12-week combination of the HCV NS3/4A inhibitor grazoprevir/NS5A inhibitor elbasvir after the surgery of HCC. Before this retreatment, his HCV RNA level was 5.6 LIU/mL. Although he achieved SVR24 by this regimen, HCC was relapsed.

### 3.3. Twelve-Week Combination of Glecaprevir/Pibrentasvir for DAA-Failure Patients

There were nine DAA-experienced patients: three HCV genotype 1b-relapsers with chronic hepatitis received HCV NS3/4A inhibitor asunaprevir/NS5A inhibitor daclatasvir; two HCV genotype 1b-patients (one is compensated cirrhosis and the other is chronic hepatitis) discontinued NS5B inhibitor sofosbuvir/NS5Ainhibitor ledipasvir due to adverse events of arrhythmia [16]; one HCV genotype 1b-relapser with chronic hepatitis received grazoprevir/elbasvir; one HCV-genotype 2b-relapser with compensated cirrhosis received sofosbuvir/ribavirin; one HCV genotype 1b-relapser with chronic hepatitis received a second DAA combination of asunaprevir/daclatasvir/NS5B inhibitor beclabuvir following the relapse after the first DAA combination of asunaprevir/daclatasvir; and one HCV genotype 1b-relapser with chronic hepatitis who received the third DAA combination of asunaprevir/daclatasvir/NS5B beclabuvir following the relapse after the second DAA combination of sofosbuvir/ledipasvir and the relapse after the first DAA combination of daclatasvir/asunaprevir. All nine patients received a 12-week combination of glecaprevir/pibrentasvir with no adverse events and achieved SVR12.

### 3.4. Combination Treatment of Glecaprevir/Pibrentasvir for Patients Undergoing Artificial Dialysis

One cirrhotic patient and five patients with chronic hepatitis were treated with 12 or 8-week combinations of glecaprevir/pibrentasvir, respectively, and all six patients achieved SVR12 with no severe adverse events (Figure 3).

As these six patients took at least eight drugs, polypharmacy seemed common among this group of patients. Careful attention should be paid to the drug-drug interaction under the combination treatment of glecaprevir/pibrentasvir. One patient complained of her pruritus, but it was improved by the oral administration of nalfurafine hydrochloride. Of interest, 4 (66.7%) of the 6 patients took nalfurafine hydrochloride during the combination treatment of glecaprevir/pibrentasvir (Table 3).

One and five patients were undergoing peritoneal dialysis and hemodialysis, respectively. Thus, 8 or 12-week combination of glecaprevir/pibrentasvir could safely treat patients undergoing artificial dialysis, irrespective of a type of artificial dialysis, and achieve higher SVR rates (Figure 3).

## 4. Discussion

In this study, real-world data from the northern part of Tokyo indicates that an 8- or 12-week combination treatment of glecaprevir/pibrentasvir could lead to 99.0% (101/102) SVR rates in HCV-infected patients with various background characteristics. Three patients discontinued the treatment because of adverse events: one had a cerebral hemorrhage, and two had hyperbilirubinemia. We assessed the cerebral hemorrhage as being unlikely related to DAAs. Two patients with compensated cirrhosis had grade 2 elevations (i.e., >1.5–3.0× upper limit of normal) in total bilirubin levels; all elevations involved direct bilirubin and were not accompanied by elevation in alanine aminotransaminase (ALT) levels. Thus, the 8- or 12-week combination treatment of glecaprevir/pibrentasvir could achieve higher SVR rates. However, clinicians should pay attention to adverse events during treatment.

Serious adverse events associated with glecaprevir/pibrentasvir treatment were low rates (2.9% (3/102)), similar to those observed in the NS5B nucleotide polymerase inhibitor-including regimen of sofosbuvir/ribavirin (1.2% (1/86); *p* = 0.400) in our hospital [6]. The combination of glecaprevir/pibrentasvir is a contraindicated regimen in the presence of advanced decompensated cirrhosis [17,18,19]. Therefore, careful attention should also be paid to the elevation of bilirubin levels in patients with cirrhosis.

We observed HCV RNA relapse at week 12 after the 8-week combination treatment of glecaprevir/pibrentasvir in one treatment-naïve patient with HCV genotype 1b and chronic hepatitis (Table 2). According to the Japanese national insurance system, 8-week or 12-week combination treatment of glecaprevir/pibrentasvir was given for DAA-naïve or DAA-experienced patients, respectively. In the United States, the 8-week combination of glecaprevir/pibrentasvir or the 12-week combination of sofosbuvir/NS5A inhibitor velpatasvir is recommended for treatment-naïve persons without liver cirrhosis, regardless of the HCV genotype [18]. A shorter duration of treatment may be desirable to reduce the cost of treatment and the occurrence rate of adverse events for DAA-treatment-naïve patients with HCV infection [20]. Careful post-treatment follow-up of patients with or without cirrhosis should also be performed for the monitoring of HCC occurrence [21]. 

Previous study demonstrated that 2 out of 2 (100%) patients who had P32 deletion in HCV NS5A at baseline, experienced virologic failure [22]. P32 deletion in the HCV genotype 1 NS5A confers > 1000-fold resistance to piblentasvir [23]. In Japan, the 24-week combination retreatment of sofosbuvir/velpatasvir plus ribavirin are recommended for HCV-infected patients with virologic failure who had both HCV genotype 1b infection and P32 deletion in the HCV NS5A region at baseline [11]. In our hospital, no HCV genotype 1-infected patients with virologic failure and this mutation, were found. Before the retreatment of DAA-failure patients, we excluded patients with virologic failure who had both HCV genotype 1b infection and P32 deletion in the HCV NS5A region at baseline [11]. After that, we successfully retreated nine patients with DAA failure. HCV genotype 1b with P32 deletion in the HCV NS5A region is more resistant to HCV NS5A inhibitors in vitro and in vivo [24,25,26]. We reconfirmed the previous report that glecaprevir/piblentasvir was effective for HCV-infected patients who failed to achieve an SVR after prior DAA therapies except in those with HCV genotype 1b carrying NS5A-P32 deletion mutation [25]. Therefore, a 12-week combination of glecaprevir/pibrentasvir could successfully retreat patients who had neither HCV genotype 1b infection nor P32 deletion in the HCV NS5A region at baseline [11].

Other studies showed that the combination treatment of glecaprevir/pibrentasvir is less effective in subjects with HCV genotype 3 [7,27], although the 3 patients with HCV genotype 3 responded well in the present study. However, the small number of subjects limits this observation and additional studies are needed in HCV genotype 3 patient population.

We also demonstrated higher efficacy and safety for the combination treatment of glecaprevir/pibrentasvir in six patients with artificial dialysis. In general, patients with artificial dialysis have polypharmacy, and attention should be paid to the interaction between these drugs and DAAs in the combination treatment of NS3/4A inhibitors/NS5A inhibitors (Table 3). Pruritus may be associated with hemodialysis. Of interest, 4 of these 6 patients took nalfurafine hydrochloride for their pruritus. Pruritus was the most frequent adverse event (30.5%) among patients who had severe renal impairment and received the combination treatment of glecaprevir/pibrentasvir [28]. HCV infection is common in hemodialysis units [29]. Our data supported several HCV guidelines that the combination treatment of glecaprevir/pibrentasvir is highly effective for patients on dialysis [16,17,18].

We observed one treatment relapse after 8 weeks of combination treatment with glecaprevir/pibrentasvir. HCC occurred ~2 years after the commencement of 8-week combination treatment with glecaprevir/pibrentasvir. It was reported that the existence of HCC could be associated with DAA treatment failure [30].

In Japan, the national health insurance system has approved the combination treatment of glecaprevir/pibrentasvir for HCV-infected patients with chronic hepatitis or Child-Pugh A cirrhosis. So, we excluded HCV-infected patients with Child-Pugh B or C cirrhosis from this study. The Japanese national health insurance system has approved the 12-week combination treatment of sofosbuvir/velpatasvir for HCV-infected patients with Child-Pugh B or C cirrhosis [31].

Prophylactic HCV vaccines are under development, although they will be needed for successful global elimination of HCV infection [32,33]. Therefore, we should have several options to eradicate this virus. In the present study, approximately one third of the patient population had compensated cirrhosis, 91.2% were HCV DAA-treatment naïve, six were on dialysis, and the genotypes were 1 (52.9%), 2 (44.1%) and 3 (0.3%). Treatment outcomes were excellent with only one subject failing to achieve SVR. 

Overall, the data provided is strong in showing that the non-clinical trial use of glecaprevir/pibrentasvir therapy is highly effective. The real-world clinical practice use of 8-week glecaprevir/pibrentasvir in treatment-naïve patients with compensated cirrhosis demonstrated that only one patient (0.5%) experienced virologic failure and treatment was well tolerated [34]. Our results are also consistent with those of the phase 3 trial from other countries [7,10,27]. There are several limitations, which include the retrospective nature of the work, the lack of a comparator group, and the exclusion of subjects with HIV and/or Child-Pugh B cirrhosis. Nevertheless, this study is reassuring and provides another real world study supporting the safety and efficacy of this combination HCV antiviral therapy [34].

## 5. Conclusions

In conclusion, the combination treatment of glecaprevir/pibrentasvir had a high efficacy and an acceptable safety profile for real-world HCV patients in the northern part of Tokyo, Japan. Treatment adherence was high regardless of the condition of the patients.

## Figures and Tables

**Figure 1 jcm-10-05529-f001:**
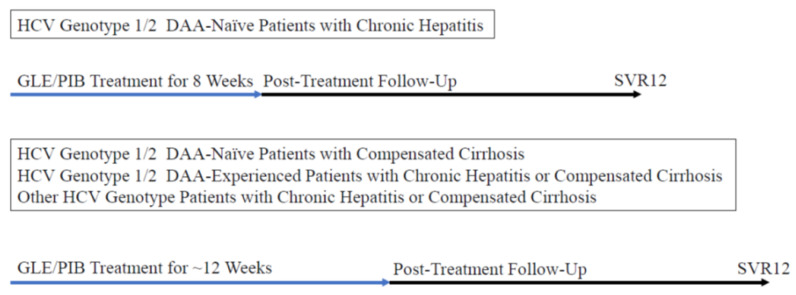
Treatment regimens in the groups of various patients. DAA, direct-acting antiviral; GLE/PIB, glecaprevir/pibrentasvir; SVR12, sustained virologic response at 12 weeks after the end of treatment.

**Figure 2 jcm-10-05529-f002:**
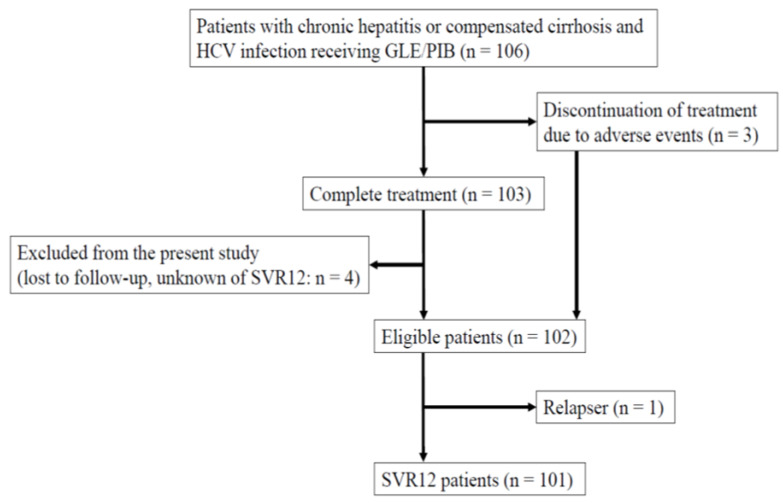
Study profile of this retrospective study.

**Figure 3 jcm-10-05529-f003:**
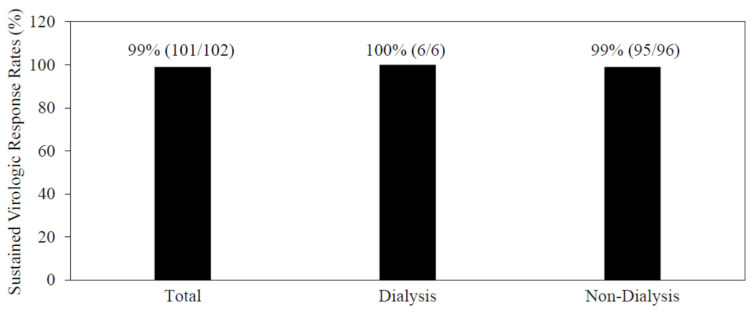
Higher sustained virologic response rates of combination treatment of glecaprevir/pibrentasvir for patients with or without dialysis in the present study.

**Table 1 jcm-10-05529-t001:** Baseline characteristics of 102 patients in the study.

Characteristics	All (*n* = 102)
Age (years)	62.7 ± 12.1
Gender (male/female)	41/61
Interferon (naïve/experienced)	88/14
DAAs (naïve/experienced)	93/9
HCV genotypes (1/2/3)	54/45/3
Pretreatment HCV RNA (LIU/mL)	6.0 ± 1.2
Body weight (kg)	58.0 ± 12.9
Body length (m)	1.60 ± 0.10
History of HCC (+/−)	5/97
Chronic hepatitis/cirrhosis	74/28
Liver stiffness (kPa)	9.9 ± 7.9
AST (IU/L)	50.7 ± 30.4
ALT (IU/L)	51.4 ± 39.1
Hemoglobin (g/dL)	13.5 ± 1.6
Platelets (×10^4^/μL)	17.6 ± 6.3
eGFR (mL/min/1.73 m^2^)	67.6 ± 26.8

HCV, hepatitis C virus; HCC, hepatocellular carcinoma; AST, aspartate aminotransferase; ALT, alanine aminotransferase; eGFR, estimated glomerular filtration rate.

**Table 2 jcm-10-05529-t002:** Baseline characteristics of a relapser after 8 weeks of glecaprevir/pibrentasvir.

Characteristics	A Relapser at Week 12 after Treatment
Age (years)	65
Gender	Male
Interferon	Naive
Interferon-free DAAs	Naive
HCV genotypes	1b
Pretreatment HCV RNA (LIU/mL)	5.4
Body weight (kg)	51
Body length (m)	1.58
History of HCC	No
Chronic hepatitis or cirrhosis	Chronic hepatitis
Liver stiffness (kPa)	7.9
AST (IU/L)	91
ALT (IU/L)	80
Hemoglobin (g/dL)	14.1
Platelets (×10^4^/μL)	23.8
eGFR (mL/min/1.73 m^2^)	64.4
Adherence > 80%	Yes
* NS5A-L31	Wild
* NS5A-Y93	Wild

DAA, direct-acting antivirals; HCV, hepatitis C virus; HCC, hepatocellular carcinoma; AST, aspartate aminotransferase; ALT, alanine aminotransferase; eGFR, estimated glomerular filtration rate. * Resistance-associated substitutions (NS5A-L31 and Y93) after treatment-relapse were determined by direct-sequence methods.

**Table 3 jcm-10-05529-t003:** Characteristics of six patients undergoing artificial dialysis with glecaprevir/pibrentasvir treatment.

Characteristics	No. 1	No. 2	No. 3	No. 4	No. 5	No. 6
Age (years)	82	84	55	57	56	64
Gender	Male	Female	Male	Male	Male	Male
Interferon	Experienced	Naive	Naive	Naive	Naive	Naive
Interferon-free DAAs	Naive	Naive	Naive	Naive	Naive	Naive
HCV GTs	1b	1b	2b	2a	2	2b
Pretreatment HCV RNA (LIU/mL)	6.8	6.3	4.8	3.9	3.3	5.3
Body weight (kg)	58.4	36.5	88.4	67.5	71.9	64.5
Body length (m)	1.60	1.48	1.73	169	1.79	1.64
History of HCC	No	No	No	No	No	No
CH or LC	LC	CH	CH	CH	CH	CH
Liver stiffness (kPa)	13.6	8.3	11.5	11.8	6.1	4.4
AST (IU/L)	50	22	72	27	15	16
ALT (IU/L)	63	10	80	24	17	13
Hemoglobin (g/dL)	14.2	8.8	9.2	10.4	13.7	10.4
Platelets (x 104/μL)	18.2	14.3	16.1	17.3	18.6	15.5
eGFR (mL/min/1.73 m^2^)	7.5	7.9	3.8	5	5	6.4
Type of dialysis	HD	HD	PD	HD	HD	HD
Duration of dialysis (years)	0.5	3.5	2	4.5	5	7
DM	No	Yes	Yes	Yes	Yes	Yes
Number of drugs under treatment	8	13	17	12	10	15
Nalfurafine hydrochloride	Yes	Yes	Yes	Yes	No	No

DAA, direct-acting antivirals; HCV, hepatitis C virus; GT, genotype; HCC, hepatocellular carcinoma; CH, chronic hepatitis; LC, liver cirrhosis; AST, aspartate aminotransferase; ALT, alanine aminotransferase; eGFR, estimated glomerular filtration rate; HD, hemodialysis; PD, peritoneal dialysis; DM, diabetes mellitus.

## Data Availability

All data underlying this article are available in this article.

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
