# Peer review of "Efficacy of Glecaprevir/Pibrentasvir for Real-World HCV Infected Patients in the Northern Part of Tokyo, Japan"

_jcm, 2021, doi:10.3390/jcm10235529_

Round 1

Reviewer 1 Report

The manuscript presents the results of using glecaprevir/pibrentasvir in patients with HCV infection from Japan. It is a real-world study based on a retrospective analysis of adult patients with HCV. One hundred six patients were initially y included, but the authors analyzed 102.

The paper is, in general, well written, with good use of the English language. Some improvements must be made, starting with the abstract. Here, there is an editing error in line 17 (piblentasvir?). Also, better to have a structured abstract. In the end, the SVR rate of 99% is repeated (both in lines 22 and 28).

In the Study design, it would be better to clearly describe the inclusion criteria before the exclusion criteria. Also, better to present in line 54 when 8 or 12 weeks were indicated in patients DAA-naive.

In the presentation of the results, the paragraph in lines 155-157 could be moved to discussions and debated. The same for 173-175.

Data presented in Figure 3 could be presented in a single sentence as the difference between 99 and 100% is not so relevant to be presented in a figure.  

In the Conclusions paragraph, the authors should only present conclusions based on their results and not add other information based on cited references. Best to move the sentences about the prophylactic vaccines in the Discussions. Also, the comparison with other trials should be discussed in the previous paragraph and not include citations in conclusions. Here there should be only a conclusion regarding this consistency. 

Author Response

Response to reviewer 1

Thank you very much for your invaluable comments.

Response to your comments: “The paper is, in general, well written, with good use of the English language. Some improvements must be made, starting with the abstract. Here, there is an editing error in line 17 (piblentasvir?). Also, better to have a structured abstract. In the end, the SVR rate of 99% is repeated (both in lines 22 and 28).”

Thank you very much for your invaluable comments.

I agree with you. Accordingly, we extensively revised our manuscript as follows.

In abstract section, lines 17 - 19, page 1

The objective of our study was to examine the effects of the HCV nonstructural protein (NS) 3/4A inhibitor glecaprevir/NS5A inhibitor pibrentasvir on real-world HCV patients in the northern part of Tokyo, Japan. Although one hundred six patients were consecutively included, total 102 HCV-infected patients with chronic hepatitis or compensated cirrhosis,…

In abstract section, line 28, page 1

…experienced patients. Glecaprevir/pibrentasvir had a high efficacy and an acceptable safety profile for real-world HCV patients in a single hospital in Japan.

Response to your comments: “In the Study design, it would be better to clearly describe the inclusion criteria before the exclusion criteria. Also, better to present in line 54 when 8 or 12 weeks were indicated in patients DAA-naive.”

Thank you very much for your invaluable comments.

I agree with you. Accordingly, we extensively revised our manuscript as follows.

In Patients and Methods section, lines 55-59, page 2

…2017 to August 31, 2019. One hundred six patients were initially included. Eligible patients were 20 years of age and older and having chronic hepatitis or compensated cirrhosis (Child-Pugh A cirrhosis). Eight or 12-week combination treatment of glecaprevir/pibrentasvir was given in DAA-naïve patients, and 12-week combination treatment of glecaprevir/pibrentasvir was given in DAA-experienced patients (Figure 1).

Response to your comments: “In the presentation of the results, the paragraph in lines 155-157 could be moved to discussions and debated. The same for 173-175.”

Thank you very much for your invaluable comments. I agree with you. Accordingly, we revised our manuscript.

Response to your comments: “Data presented in Figure 3 could be presented in a single sentence as the difference between 99 and 100% is not so relevant to be presented in a figure.”

Thank you very much for your invaluable comments. I agree with you. Accordingly, we revised Figure 3 legends of our manuscript.

Response to your comments: “In the Conclusions paragraph, the authors should only present conclusions based on their results and not add other information based on cited references. Best to move the sentences about the prophylactic vaccines in the Discussions. Also, the comparison with other trials should be discussed in the previous paragraph and not include citations in conclusions. Here there should be only a conclusion regarding this consistency.”

Thank you very much for your invaluable comments. I agree with you. Accordingly, we revised our manuscript.

Reviewer 2 Report

This manuscript describes a retrospective analysis of glecaprevir/pibrentasvir (mavyret) treatment of 106 consecutive patients with chronic HCV infection in a “real world” clinical setting located in Tokyo.  Approximately one third of the patient population had compensated cirrhosis, 90% were HCV treatment naïve, six were on dialysis, and the genotypes were 1 (53%), 2 (44%) and 3 (3%).  Treatment outcomes were excellent with only one subject failing to achieve SVR (as measured by negative HCV RNA at 12 weeks post therapy.  Overall, the data provide strong showing that non-clinical trial use of glecaprevir/pibrentasvir therapy is highly effective.  There are several questions the authors should consider:

  1. Subjects with genotype 1b and the P32 deletion in HCV NS5a were excluded from the analysis. No description of any subjects with these findings is provided.  Were any patients considered for treatment but excluded for this NS5A deletion/genotype?  This should be noted.
  2. Child B or C cirrhosis was exclusionary. Were any patients considered for therapy who were excluded for this, and if so, how many?
  3. Of the 6 subjects on dialysis, the type of dialysis is noted in Table 3; however, this should be mentioned in the results section as well.
  4. Cirrhosis was based on elastography with a score of > 12 considered cirrhosis, or if there was evidence of varices on EGD, CT or MRI. Compensated cirrhosis presumably had Child-Pugh A.
  5. The one relapse patient is described. Is the RNA value in Table 2 (5.4 LIU/mL) pre-treatment or at 12 weeks?  Was pre-treatment, was the HCV RNA value determined at 12 weeks post-Rx and positive?  The subsequent treatment 2 years later for 12 weeks is interesting.
  6. It would seem that the comment on lines 223-226 are not confirmed in the study, unless there are unreported data showing that there were subjects with gt 1b and P32 deletion who were treated and failed to achieve SVR.
  7. The wording on line 227 could be clarified. It might be better to simply state as noted in line 229 that other studies have shown that mavyret therapy is less effective in subjects with gt 3, although the 3 gt 3 patients studied here responded well.  However, the small number of subjects limits this observation and additional studies are needed in this patient population.
  8. Finally, the paragraph starting at line 242 again raises the question of timing of HCV RNA detection/levels in the one subject who relapsed. More clear description of HCV RNA testing times and results would be helpful in interpreting this patient’s course.
  9. There is at least one real world study of glecaprevir/pibrentasvir (Adv Ther 2020, 27:4033, 4042). It would be good for the authors to compare their study with this published study to provide additional information for the generalizability of their results. 

Overall, the study is clearly written and supports high efficacy of mavyret in chronic HCV infection.  Limitations include the retrospective nature of the work, the lack of a comparator group, and the exclusion of subjects with HIV and/or Child Pugh B cirrhosis.  Nevertheless, the study is reassuring and provides another real world study supporting the safety and efficacy of this combination HCV antiviral therapy.

Author Response

Response to reviewer 2

Thank you very much for your invaluable comments.

Response to your comment 1: “Subjects with genotype 1b and the P32 deletion in HCV NS5a were excluded from the analysis. No description of any subjects with these findings is provided.  Were any patients considered for treatment but excluded for this NS5A deletion/genotype?  This should be noted.”

Thank you very much for your invaluable comments.

I agree with you. Accordingly, we extensively revised our manuscript as follows.

In Discussion section, line 235, page 8 – line 251, page 9,

Previous study demonstrated that two of 2 (100%) patients who had P32 deletion in HCV NS5A at baseline, experienced virologic failure [22]. P32 deletion in HCV genotype 1 NS5A confers > 1000-fold resistance to piblentasvir [23]. In Japan, the 24-week combination retreatment of sofosbuvir/velpatasvir plus ribavirin are recommended for HCV-infected patients with virologic failure who had both HCV genotype 1b infection and P32 deletion in the HCV NS5A region at baseline [24]. In our hospital, no HCV genotype 1-infected patients with virologic failure and this mutation, were found. Before retreatment of DAA-failure patients, we excluded patients with virologic failure who had both HCV genotype 1b infection and P32 deletion in the HCV NS5A region at baseline [11]. After that, we successfully retreated nine patients with DAA failure. HCV genotype 1b with P32 deletion in the HCV NS5A region is more resistant to HCV NS5A inhibitors in vitro and in vivo [25-27]. We reconfirmed the previous report that glecaprevir/piblentasvir was effective for HCV-infected patients who failed to achieve an SVR after prior DAA therapies except in those with HCV genotype 1b carrying NS5A-P32 deletion mutation [26]. Therefore, a 12-week combination of glecaprevir/pibrentasvir could successfully retreat patients who had neither HCV genotype 1b infection nor P32 deletion in the HCV NS5A region at baseline [11].

Response to your comment 2: “Child B or C cirrhosis was exclusionary. Were any patients considered for therapy who were excluded for this, and if so, how many?”

Thank you very much for your invaluable comments.

I agree with you. We have treated 10 HCV-infected patients with Child-Pugh B or C cirrhosis by the 12-week combination treatment of sofosbuvir/velpatasvir. Accordingly, we extensively revised our manuscript as follows.

In Discussion section, lines 272 - 277, page 9,

In Japan, national health insurance system has approved the combination treatment of glecaprevir/pibrentasvir for HCV-infected patients with chronic hepatitis or Child-Pugh A cirrhosis. So, we excluded HCV-infected patients with Child-Pugh B or C cirrhosis from this study. Japanese national health insurance system has approved the 12-week combination treatment of sofosbuvir/velpatasvir for HCV-infected patients with Child-Pugh B or C cirrhosis [32].

Response to your comment 3: “Of the 6 subjects on dialysis, the type of dialysis is noted in Table 3; however, this should be mentioned in the results section as well.”

Thank you very much for your invaluable comments.

I agree with you. Accordingly, we extensively revised our manuscript as follows.

In Results section, lines 194 - 197, page 7,

One and five patients were undergoing peritoneal dialysis and hemodialysis, respectively. Thus, 8 or 12-week combination of glecaprevir/pibrentasvir could safely treat patients undergoing artificial dialysis, irrespective of a type of artificial dialysis, and achieve higher SVR rates (Figure 3).

Response to your comment 4: “Cirrhosis was based on elastography with a score of > 12 considered cirrhosis, or if there was evidence of varices on EGD, CT or MRI. Compensated cirrhosis presumably had Child-Pugh A.”

Thank you very much for your invaluable comments.

I agree with you. Accordingly, we extensively revised our manuscript as follows.

In Patients and Methods section, lines 90 - 100, page 3,

2.4. Assessment of advanced liver fibrosis and diagnosis of cirrhosis and HCC

Ultrasonography and hepatic transient elastography on a FibroScan 502 with an M probe (Echosens, Paris, France) were performed. In general, liver stiffness equal to or more than 12.0 kPa or sign of cirrhosis was considered liver cirrhosis. Sign of cirrhosis was the existence of varices in the esophagus and/or stomach on upper gastrointestinal endoscopy or the existence of compatible findings of liver cirrhosis in computed tomography (CT) scanning or magnetic resonance imaging (MRI). In this study, Child-Pugh A cirrhosis was defined as compensated cirrhosis. In general, HCV-infected patients were followed up through an HCC surveillance program based on ultrasonography evaluations with or without tumor markers/CT/MRI at least every six months [6].

Response to your comment 5: “The one relapse patient is described. Is the RNA value in Table 2 (5.4 LIU/mL) pre-treatment or at 12 weeks? Was pre-treatment, was the HCV RNA value determined at 12 weeks post-Rx and positive? The subsequent treatment 2 years later for 12 weeks is interesting.”

Thank you very much for your invaluable comments.

I agree with you. Accordingly, we revised Table 2 and extensively revised our manuscript as follows.

In Results section, lines 151 - 160, page 5,

The characteristics of one relapse patient is shown in Table 2. In this patient, HCV RNA was relapsed after 12 weeks of the end of treatment. At this time, his HCV RNA level was 5.2 LIU/mL. He has stopped coming to our outpatient clinic due to his circumstance for two years. He received hepatic resection for HCC ~2 years after the commencement of 8-week combination treatment of glecaprevir/pibrentasvir. HCV RNA level was 5.5 LIU/mL before his surgery. Histological evaluation of non-HCC liver revealed no existence of liver cirrhosis. Due to his severe heart disease, he was retreated with the 12-week combination of the HCV NS3/4A inhibitor grazoprevir/NS5A inhibitor elbasvir after the surgery of HCC. Before this retreatment, his HCV RNA level was 5.6 LIU/mL. Although he achieved SVR24 by this regimen, HCC was relapsed.

Response to your comment 6: “It would seem that the comment on lines 223-226 are not confirmed in the study, unless there are unreported data showing that there were subjects with gt 1b and P32 deletion who were treated and failed to achieve SVR.”

Thank you very much for your invaluable comments.

I agree with you. Accordingly, we extensively revised our manuscript as follows.

In Results section, line 235, page 8 - 251, page 9,

Previous study demonstrated that two of 2 (100%) patients who had P32 deletion in HCV NS5A at baseline, experienced virologic failure [22]. P32 deletion in HCV genotype 1 NS5A confers > 1000-fold resistance to piblentasvir [23]. In Japan, the 24-week combination retreatment of sofosbuvir/velpatasvir plus ribavirin are recommended for HCV-infected patients with virologic failure who had both HCV genotype 1b infection and P32 deletion in the HCV NS5A region at baseline [24]. In our hospital, no HCV genotype 1-infected patients with virologic failure and this mutation, were found. Before retreatment of DAA-failure patients, we excluded patients with virologic failure who had both HCV genotype 1b infection and P32 deletion in the HCV NS5A region at baseline [11]. After that, we successfully retreated nine patients with DAA failure. HCV genotype 1b with P32 deletion in the HCV NS5A region is more resistant to HCV NS5A inhibitors in vitro and in vivo [25-27]. We reconfirmed the previous report that glecaprevir/piblentasvir was effective for HCV-infected patients who failed to achieve an SVR after prior DAA therapies except in those with HCV genotype 1b carrying NS5A-P32 deletion mutation [26]. Therefore, a 12-week combination of glecaprevir/pibrentasvir could successfully retreat patients who had neither HCV genotype 1b infection nor P32 deletion in the HCV NS5A region at baseline [11].

Response to your comment 7: “The wording on line 227 could be clarified. It might be better to simply state as noted in line 229 that other studies have shown that mavyret therapy is less effective in subjects with gt 3, although the 3 gt 3 patients studied here responded well.  However, the small number of subjects limits this observation and additional studies are needed in this patient population.”

Thank you very much for your invaluable comments.

I agree with you. Accordingly, we extensively revised our manuscript as follows.

In Results section, lines 252 - 256, page 9,

Other studies showed that the combination treatment of glecaprevir/pibrentasvir is less effective in subjects with HCV genotype 3 [7, 28], although the 3 patients with HCV genotype 3 responded well in the present study. However, the small number of subjects limits this observation and additional studies are needed in HCV genotype 3 patient population.

Response to your comment 8: “Finally, the paragraph starting at line 242 again raises the question of timing of HCV RNA detection/levels in the one subject who relapsed. More clear description of HCV RNA testing times and results would be helpful in interpreting this patient’s course.”

Thank you very much for your invaluable comments.

I agree with you. Accordingly, we extensively revised our manuscript as follows.

In Patients and Methods section, lines 81 - 84, page 12,

Serum HCV RNA levels were measured by COBAS TaqMan assay (Roch Diagnostics, Tokyo Japan) with detection limits of ~1.2 LIU/mL. At least, HCV RNA levels were determined at pre-treatment, at the end of treatment and after 12 weeks of the end of treatment. SVR12 was used as the SVR to evaluate the virological response. Virus clearance was…

Response to your comment 9: “There is at least one real world study of glecaprevir/pibrentasvir (Adv Ther 2020, 27:4033, 4042). It would be good for the authors to compare their study with this published study to provide additional information for the generalizability of their results.”

Thank you very much for your invaluable comments.

I agree with you. Accordingly, we extensively revised our manuscript as follows.

In Discussion section, line 284, page 9 – line 293, page 10,

Overall, the data provide strong showing that non-clinical trial use of glecaprevir/pibrentasvir therapy is highly effective. Real-world clinical practice use of 8-week glecaprevir/pibrentasvir in treatment-naïve patients with compensated cirrhosis demonstrated only one patient (0.5%) experienced virologic failure and treatment was well tolerated [35]. Our results are also consistent with those of the phase 3 trial from other countries [7, 10, 28]. There are several limitations, which include the retrospective nature of the work, the lack of a comparator group, and the exclusion of subjects with HIV and/or Child-Pugh B cirrhosis. Nevertheless, this study is reassuring and provides another real world study supporting the safety and efficacy of this combination HCV antiviral therapy.
